# Broadband Dual-Polarized Loop Cross-Dipole Antenna for 5G Base Station Applications

**Chenyang Liao, Bin Wang \*** 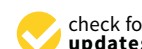**, Congcong Zhu, Honggang Hao and Bo Yin**

College of Electronic Engineering, Chongqing University of Posts and Telecommunications, Chongqing 400065, China; s180431048@stu.cqupt.edu.cn (C.L.); s170401004@stu.cqupt.edu.cn (C.Z.); haohg@cqupt.edu.cn (H.H.); yinbo@cqupt.edu.cn (B.Y.)

**\*** Correspondence: wangbin1@cqupt.edu.cn

**Abstract:** A broadband dual-polarized base station antenna is proposed in this paper. The antenna consists of loop cross-dipoles, Y-shaped coupling feeding lines, and a metal box reflector. An equivalent circuit model including a signal flow diagram is established to analyze the mechanism of the proposed antenna in detail. Moreover, the Y-shaped coupling feeding lines are introduced to control the coupling with the antenna to achieve broadband and good impedance matching. The prototype of the antenna is fabricated and measured. The measured results show that the antenna with simple structures can operate at the band of 3.2–5.22 GHz (48%) with high port-to-port isolation (35 dB) and stable gain (9 ± 1 dBi). The measured results show good agreement with simulated results, especially in cross-polarization discrimination ratio (>27 dB) and the half power beam width (61° ± 3° at the E-plane, 68° ± 3° at the H-plane). In summary, the proposed antenna could be a good candidate for 5G sub-6 GHz base station applications.

**Keywords:** base station antenna; 5G; broadband; high isolation

## 1. Introduction

Recently, 5G communication systems have received widespread attention due to their high capacity, low latency, and high data rate. The sub-6 GHz band (450 MHz–6.0 GHz) like 3.3–3.6 GHz and 4.8–5.0 GHz was announced by the Chinese Ministry of Industry and Information Technology (MIIT) in 2017 [1], 3.4–3.8 GHz in Europe and 3.4–3.7 GHz in Korea [2], which will be widely used in 5G communication systems all over the world. For the performances of base station antenna, there are higher demands. Especially in bandwidth, isolation among the ports, and stable radiation pattern. However, one of the greatest difficulties is that broadband, high port-to-port isolation, and stable radiation patterns need to be achieved simultaneously in a limited space. The limited space enhances the coupling between the antennas. Therefore, the current 5G antenna array still needs to take measures (such as decoupling structures, dielectric resonant cavities) to ensure stable patterns. Moreover, in order to avoid the influences of multipath fading and increase the capacity of the communication system, dual-polarization antennas [3] have been widely used in base station.

There are many different structures of broadband dual-polarized antennas designed for base stations in recent years, such as patch antennas, electromagnetic dipole antennas, folded dipole antennas, and cross-dipole antennas. The patch antenna is very popular because of its simple structure and low cost [4–6], but it covers a narrow bandwidth [7], relatively. A high gain stacked patch antenna was proposed in [8], which covered a bandwidth (2.31–2.81 GHz, 19%), and its gain was 11 dBi. A broadband dual-polarized stacked patch antenna with specific feeding structure was designed in [9]. The antenna covered 1.71–2.72 GHz, and the cross-polarization discrimination (XPD) was above 23 dB. By using two different feeding baluns, a patch antenna was proposed in [10], which realized the

wideband (2.2–2.6 GHz, 18%) and the high isolation (28.5 dB). In conclusion, various feeding structures and shapes of patch could be used to expand the bandwidth and increase the isolation. Unfortunately, the effects are limited.

With the increasing demand for bandwidth, traditional patch antennas are not suitable for providing wider impedance bandwidth, and the dipole antenna becomes another good choice for achieving the wideband. Then, the folded dipole can get a wide band, which had been studied since early times. A folded dipole antenna with wide bandwidth (1.71–2.17 GHz, 21.2%) and high isolation (30 dB) was proposed in [11]. Later, a wideband folded dipole antenna was developed in [12], operating from 1.69 GHz to 2.71 GHz, and the XPD was above 25 dB. Subsequently, a broadband dual-polarized folded dipole antenna was proposed in [13] with a stable radiation pattern and good impedance matching, and it is worth mentioning that the profile of antenna is lower than others which were proposed in the literature. However, the main drawback of such antennas is that the configuration of the antennas is usually complicated.

In addition to the folded dipole antenna, the electromagnetic dipole antenna with good performance in wideband and dual-polarized is also a popular choice. In order to realize the broadband, the Γ-shaped feeding trips have been applied for a dual-polarized electromagnetic dipole antenna in [14], and the antenna covered the wideband (1.58–3.74 GHz, 81.2%) with a stable radiation pattern. By employing four novel metal walls, a wideband dual-polarized electromagnetic dipole antenna was proposed in [15], the bandwidth of antenna was 1.62–2.87 GHz (56%) and the radiation performance was stable. Overall, the electromagnetic dipole base station antennas can not only cover a wideband but also realize stable radiation patterns. Nevertheless, because of the non-planar configuration, some of the properties of the electromagnetic dipole will be affected.

The mutual resonance of two pairs of cross-dipole can introduce a new resonance mode to expand the bandwidth, and it is easy to fabricate. That is why the cross dipole is widely used in base station antennas. Then, a broadband dual-polarized cross-dipole antenna was reported in [16], which operated from 1.68 GHz to 2.74 GHz (48%). However, the isolation of the antenna was only 22 dB. Fortunately, wider bandwidth and higher isolation were achieved in [17]. By researching the mutual coupling with cross-dipoles, the bandwidth from 1.69 GHz to 2.95 GHz (54%) and high isolation (34 dB) were achieved. As the research progressed, another form of cross-dipole-like loop cross-dipole (the rest is referred to as loop dipole) was proposed. With a simple feeding structure, the dual-polarized loop dipole antenna was designed in [18], which showed good performance in the band from 1.7 GHz to 2.7 GHz and a stable gain. Unfortunately, the isolation was about 25 dB. By using a loop dipole with chamfers as radiator in [19], the antenna had a wideband (1.7–2.8 GHz, 48.8%) for $S_{11} < -15$ dB and high XPD. A multi-mode loop dipole with two resonators covered the band from 1.69 GHz to 3.6 GHz (72.2%) for VSWR < 2 was proposed in [20]. Additionally, there were other forms of dipoles, like antennas proposed in [21,22]. All in all, to ensure the good performance both in electrical parameters and radiation parameters of the antenna, the loop dipole would be a good choice in these structures. In addition, such antennas can be combined with structures such as resonant cavities to form resonant cavity antennas for base station applications in 5G communications. The resonant cavity can not only improve the gain and isolation of the base station antenna, but also make the structure of antennas more compact so that it can be installed in a more limited space. In [23], a broadband polarization-reconfigurable Fabry–Perot (FP) resonator antenna was presented. By loading the FP cavity, the antenna obtained the wide band at 2.2–2.7 GHz, and realized the peak gain of 15.1 dBi. The next year, a dual-polarized antenna with compact structure was proposed for the fifth-generation base stations at 3.5 GHz. The broadband and compact structure were realized because of the independent dielectric resonant cavity in [24]. The resonant cavity was proven to suppress the coupling between antenna elements in this letter. A problem of low gain for antennas with a wider frequency band was solved by loading a dielectric resonant cavity in [25]. The antenna can operate from 6.5 GHz to 10.1 GHz, and the gain of the antenna was 12.5 dBi. In addition, there was more about

the application of resonant cavities in 5G communication systems in [26]. All things considered, it is a very meaningful research direction.

In this paper, our work focuses on the proposal of a broadband dual-polarization loop dipole antenna with simple structure and good performance for 5G base station. The impedance matching and the bandwidth have been improved and expanded by introducing Y-shaped feeding lines. The stable gain and radiation patterns have been achieved by loop dipoles and a metal box reflector. Additionally, an equivalent circuit is used to demonstrate the feasibility of proposed antenna. The measured results are in good agreement with the simulation results. In summary, it operated at the sub-6GHz band, and could be a good candidate for 5G base station applications.

## 2. Antenna Geometry

The proposed antenna is shown in Figure 1, which consists of a pair of loop dipoles, Y-shaped feeding lines, a pair of coaxial cables, four plastic columns and a metal box reflector. The loop dipoles are located at the bottom of the 0.8 mm thick FR4 substrate with a length of *W*. At the top of the FR4 substrate, there are Y-shaped feed lines and the part of other structure for feeding. In order to ensure the profile height of the prototype, plastic columns and nuts are used to support the substrate, as shown in Figure 1d.

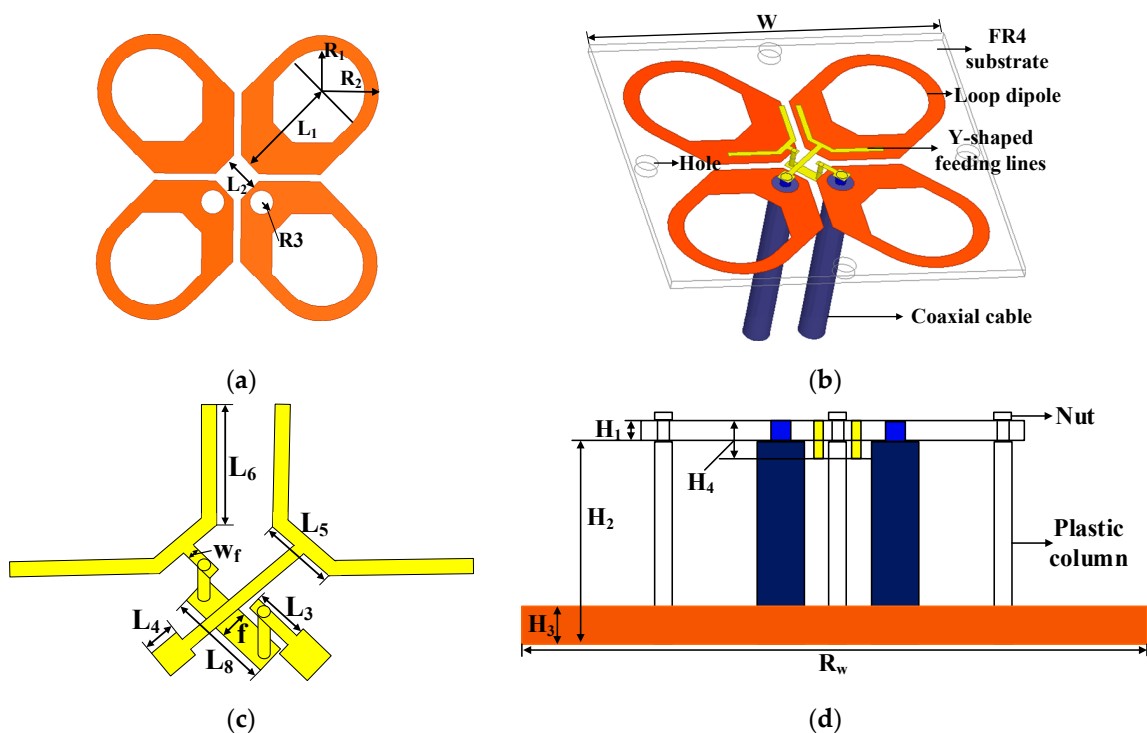

**Figure 1.** Geometry of the proposed antenna. (**a**) Top view of the radiator, (**b**) 3D view of the antenna, (**c**) perspective view of the feeding lines, and (**d**) side view of the antenna.

The outer of the two coaxial cables are connected to loop dipoles directly, while the inner of two coaxial cables are connected to the Y-shaped feed lines. The feeding structures consist of two Y-shaped feeding lines, a line with the length of $L_8$, and two metal columns with a height of $H_4$ to ensure the isolation of the two ports. The width of $L_8$ can improve impedance matching, while the length of $L_6$ and the position of Y-shaped feeding lines can change the strength of coupling so that they can affect the bandwidth and impedance matching efficiently. Additionally, a metal box reflector is used for achieving high gain and stable radiation pattern, whose size is $1.25\lambda_0 \times 1.25\lambda_0 \times 0.13\lambda_0$, and the height of profile is $0.25\lambda_0$, where $\lambda_0$ is the free space wavelength at the antenna central frequency (4.2 GHz). The optimized parameters of the proposed antenna are shown in Table 1.

**Table 1.** Parameters of the proposed antenna (units: mm).

| Parameters | Value | Parameters | Value |
|:---:|:---:|:---:|:---:|
| $R_1$ | 4.2 | $L_8$ | 3.6 |
| $R_2$ | 7.5 | $W_f$ | 0.5 |
| $R_3$ | 1.2 | $f$ | 1.0 |
| $L_1$ | 5.8 | $R_W$ | 90.0 |
| $L_2$ | 3.7 | $H_1$ | 0.8 |
| $L_3$ | 1.9 | $H_2$ | 18.0 |
| $L_4$ | 1.2 | $H_3$ | 10.0 |
| $L_5$ | 2.6 | $H_4$ | 1.6 |
| $L_6$ | 4.7 | $W$ | 32.0 |

## 3. Evolution Process and Mechanism Analysis

### 3.1. The Evolution of Antenna Structure

As a continuation of our previous work in [27], the proposed antenna has the same mechanism in achieving broadband. The evolution of the proposed antenna and the results of S11 are shown in Figures 2 and 3a. When only a pair of dipoles is fed by the outer of coaxial cable, the electromagnetic energy is coupled to another dipole by feeding lines. There is only one resonance mode in the frequency band. After loading another pair of dipoles, a new resonance mode appeared in the frequency band. Meaning that any pair of dipoles is excited to resonate, and the other pair resonates with it. Therefore, as long as the appropriate configuration is chosen and the coupling is controlled legitimately, the wide bandwidth can be achieved effectively. In addition, the second resonance mode mainly depends on the coupling, one is the coupling between the Y-shaped feeding lines and the dipole, the other is the coupling between two pairs of dipoles. The first resonance mode mainly depends on the size of the dipole itself, as shown in Figure 3b.

At the beginning, the configuration of antenna like antenna 2. In order to improve the impedance matching, the loop dipoles and optimized Y-shaped feeding lines are introducing (antenna 3 and antenna 4). Meanwhile, the coupling is increased. Unfortunately, the bandwidth becomes narrow although the matching is improved. Further, the chamfers are introduced to expand the bandwidth (the proposed antenna). The bandwidth of the proposed antenna is almost the same as that of antenna 3. Additionally, it has no effect on impedance matching. Figure 3c shows the isolation of four antennas, the optimized feeding lines can increase the isolation. The chamfers make the isolation worse, but the isolation of proposed antenna is still greater than 35 dB, which can satisfy the engineering application.

### 3.2. Mechanism Analysis

Figure 4 illustrates the current distribution on loop dipoles at different frequencies. The current amplitude reaches a maximum in a period. As the frequency increases, the current gradually concentrates on the edge of the loop dipole. The current at 3.4 GHz and 4.8 GHz is greater than that at 4 GHz and 4.2 GHz. In conclusion, the impedance matching is good at 3.4 GHz and 4.8 GHz, which will be confirmed in the following content.

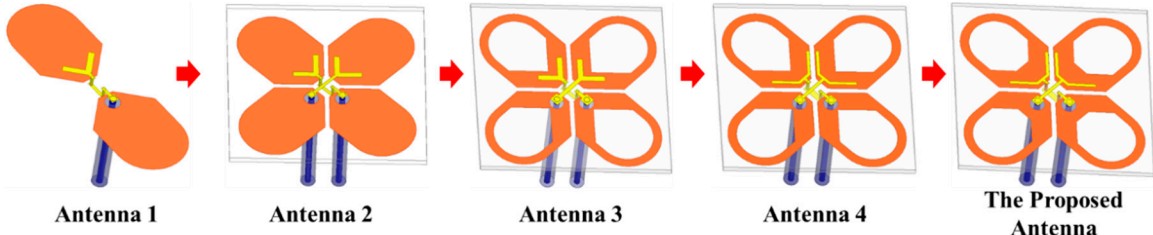

**Figure 2.** Evolution of antenna structure.

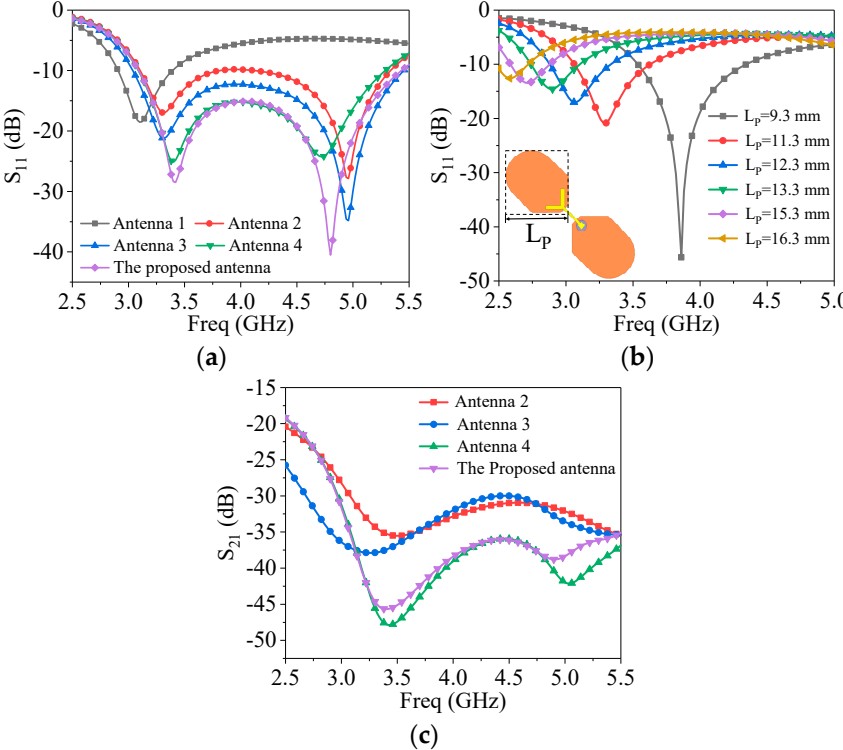

**Figure 3.** The results of the S-parameters. (**a**) The $S_{11}$ of evolution, (**b**) the $S_{11}$ of different $L_P$, and (**c**) the $S_{21}$ of evolution.

According to the description of the above principle, an approximate equivalent circuit proposed for the antenna is established to insight into the working mechanism, as shown in Figure 5. Based on the equivalent circuit model proposed in [24], the dipole antenna could be equivalent to the lumped parameter model of the following five elements, which $C_1$ and $L_1$ refer to the equivalent capacitance and inductance of the antenna at low frequency respectively, and the resistance could be ignored. The Y-shaped feeding lines are equivalent to capacitances ($C_5$, $C_6$) and inductances ($L_5$, $L_6$). To approximate the coupling effect between the dipoles, capacitances ($C_7$, $C_8$) are also introduced. Additionally, the $R_5$ is the resistance of the outer of the coaxial cable in contact with the loop dipole. Then, the values of part parameters could be calculated by Equations (1)–(6) in [28].

$$C_1 = C_2 = \frac{\pi \varepsilon_0 l}{ln(2l/r) - 1 - ln2} \tag{1}$$

$$L_1 = L_2 = \frac{1}{4p^2 f_1^2 C1} \tag{2}$$

$$C_3 = C_4 = \frac{1}{2\pi} \frac{A}{R_0^2 + A^2} \left( \frac{f_1}{f_1^2 - f_2^2} \right) \tag{3}$$

$$L_3 = L_4 = \frac{1}{4\pi^2 f_2^2 C_1} \tag{4}$$

$$R_3 = R_4 = \left( R_0^2 + A^2 \right) / R_0 \tag{5}$$

$$A = 2\pi f_1 L_0 - (1/2\pi f_1 C_0) \tag{6}$$

where $R_0 = 73\ \Omega$ and the value of $R_0$ was given in [29], $f_1$ and $f_2$ are resonant frequencies at low and high frequencies, respectively.

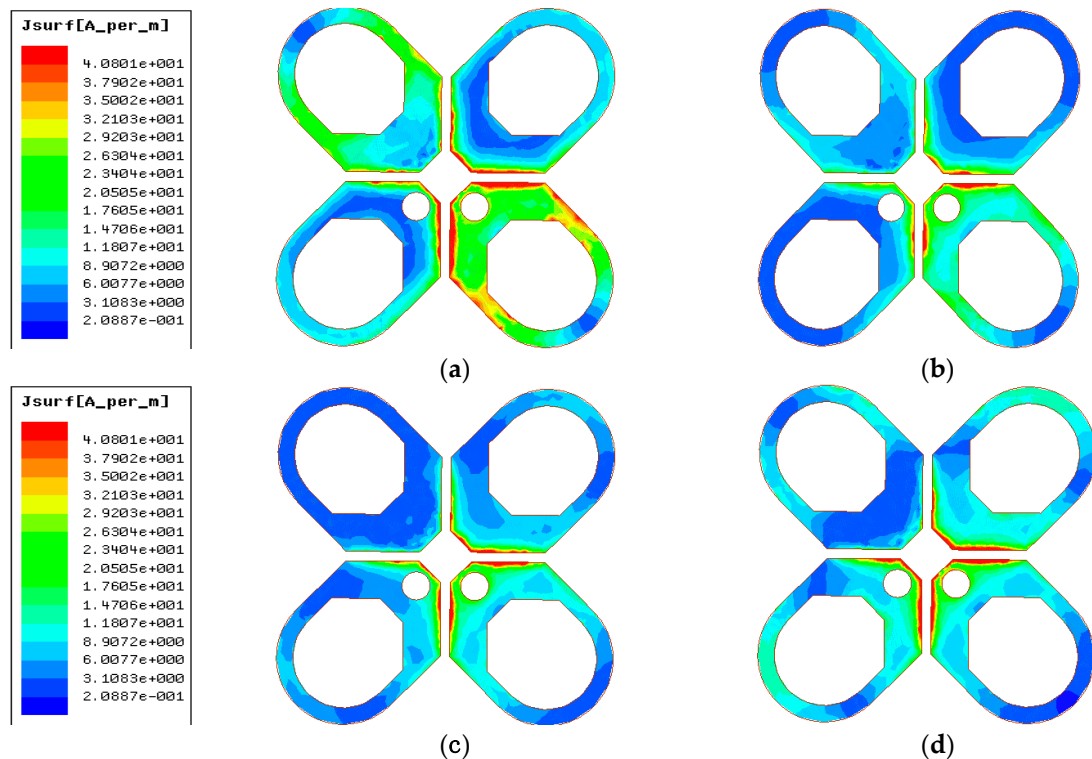

**Figure 4.** Current distribution. (**a**) 3.4 GHz, (**b**) 4 GHz, (**c**) 4.2 GHz, (**d**) 4.8 GHz.

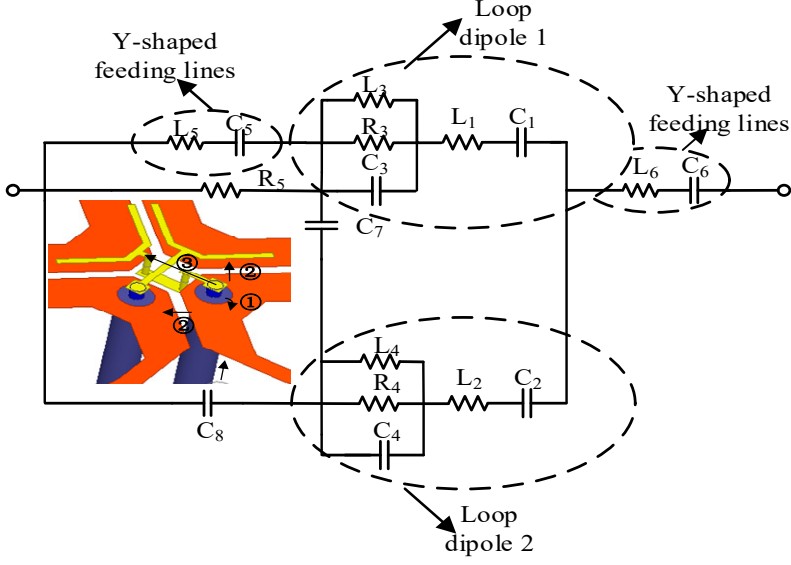

**Figure 5.** The equivalent circuit of the proposed antenna.

The electromagnetic energy is fed by the port on the left side of the equivalent circuit, and then divided into three channels. The first is that the inner of coaxial cable fed to Y-shaped feeding lines. The second is that the outer of coaxial cable fed to the loop dipole 1. The third is the loop dipole 1 coupled to the loop dipole 2. After the signal passes through the Y-shaped feeding lines, there is still electromagnetic energy coupled from the loop dipole 1 to the loop dipole 2, so $C_7$ is introduced to be equivalent to this process.

Through calculation, the initial values of each lumped element can be obtained, and the equivalent circuit is simulated and optimized in Advanced Design System software (ADS) as shown in Figure 6b.

Additionally, the simulated results in HFSS are given in Figure 6a simultaneously. Since it is not precisely equivalent, there will be some deviation between the results in ADS and those in HFSS. However, a good agreement of the variation trend at 3.2–5.2 GHz of $S_{11}$ and $S_{22}$ verifies the validity of the equivalent circuit, especially in the two resonant frequencies (about 3.4 GHz and 4.8 GHz).

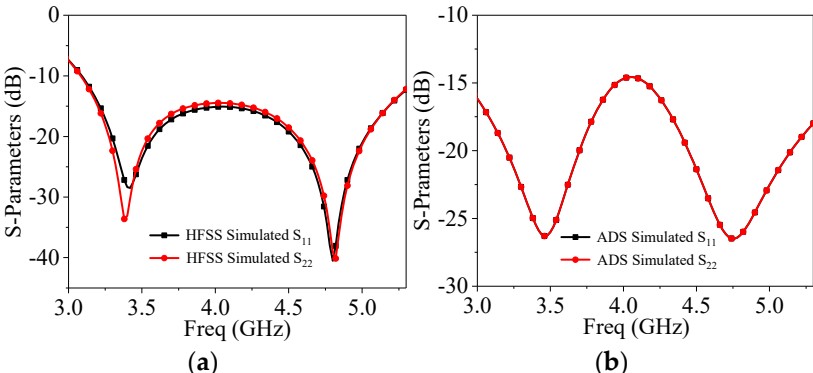

**Figure 6.** Comparison of simulation results between HFSS and ADS. (**a**) HFSS, (**b**) ADS.

## 4. Parametric Analysis

### 4.1. Y-Shaped Feeding Lines

The variation of $S_{11}$ with part of Y-shaped feeding lines parameters is listed in Figure 7. As shown in Figure 7a, by changing the length of Y-shaped feeding lines ($L_6$), the bandwidth changes significantly, meaning that the coupling strength can be optimized by changing the length of $L_6$. As $L_6$ decreases, the bandwidth gets wider, but the impedance matching gets worse. In addition, the value of $H_4$ also has a great influence on the coupling strength, which in turn affects the bandwidth. As seen from Figure 7b, the wider bandwidth can be obtained by appropriately reducing the value of $H_4$, but the impedance matching will also deteriorate accordingly. Therefore, properly control the coupling strength so that bandwidth and good impedance matching can be achieved (when $L_6 = 4.7$ mm, $H_4 = 1.6$ mm).

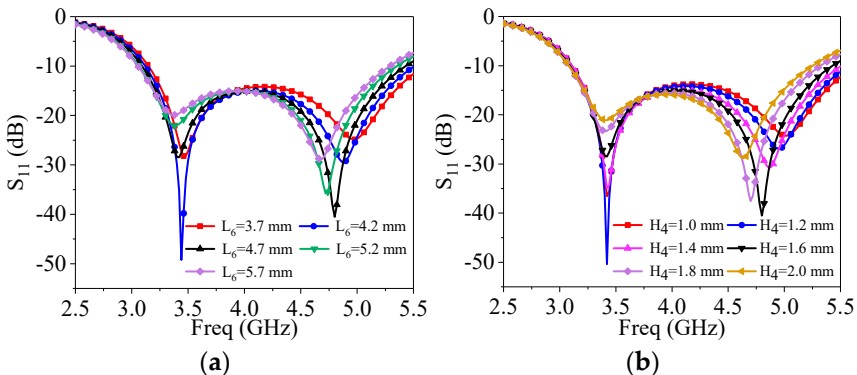

**Figure 7.** Parametric analysis of Y-shaped feeding lines. (**a**) $L_6$, (**b**) $H_4$.

### 4.2. Trepanning Analysis

Since nuts and plastic columns are introduced to ensure the profile, four holes are considered in the substrate, which shown in Figure 1b. Therefore, it is necessary to choose the suitable position of the holes. Figure 8 illustrates the simulated S-parameters for two cases of trepanning on FR4 substrate. In case 1, the holes are considered at ±45° of the substrate, while the holes are considered at the 0° and 90° of the substrate in case 2. As can be seen from Figure 8a,b, it is similar to $S_{11}$ and $S_{22}$, but $S_{21}$ is lower than that in case 2. Therefore, case 2 is introduced to the proposed antenna. In addition,

due to the limitation of the size of the plastic column, the radius of holes is determined accordingly. Nevertheless, the influence of the holes radius on the S-parameters is not discussed in this paper.

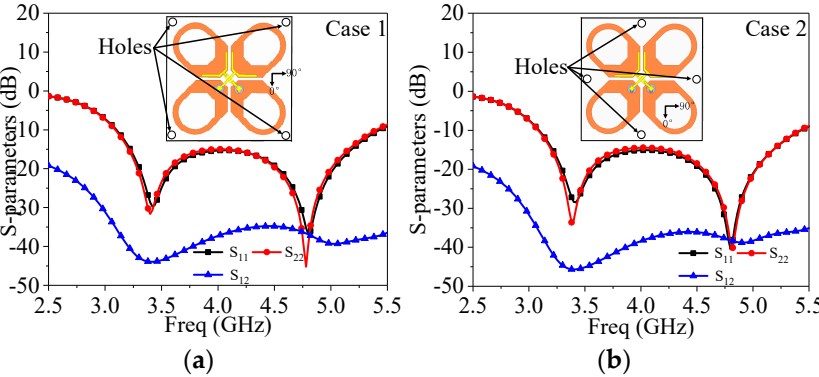

**Figure 8.** Two cases of trepanning. (**a**) Case 1, (**b**) case 2.

## 5. Result and Discussion

As shown in Figure 9, the proposed antenna is fabricated to validate the simulated results. The simulated and measured S-parameters are illustrated in Figure 10. In order to ensure the connection between the antenna and the test equipment in a microwave anechoic chamber, the length of the actually adopted coaxial cables are much longer than that of the simulated coaxial cables. Thus the loss of the coaxial cables has been considered before measuring. Then, from the Figure 10a, the simulated and measured bandwidth at port 1 is 47.6% (3.2–5.2 GHz) and 48% (3.2–5.22 GHz) with $S_{11} < -15$ dB, respectively. While at port 2, the bandwidth resulted by simulation and measurement is 47.7% (3.19–5.2 GHz) and 48.1% (3.18–5.22 GHz), respectively. At about 4 GHz, $S_{22}$ is slightly greater than −15 dB due to the impedance mismatch caused by the uneven contact surface between $L_8$ and the metal column. The measured $S_{21}$ remains under −35 dB over the band, which is slightly above the simulation result. As shown in Figure 10b, the simulated gain is from 8.46 dBi to 10.20 dBi, while the measured gain is from 8.10 dBi to 10.10 dBi. In summary, the difference between simulation and actual measurement may come from the loss of coaxial cables, machining and welding accuracy.

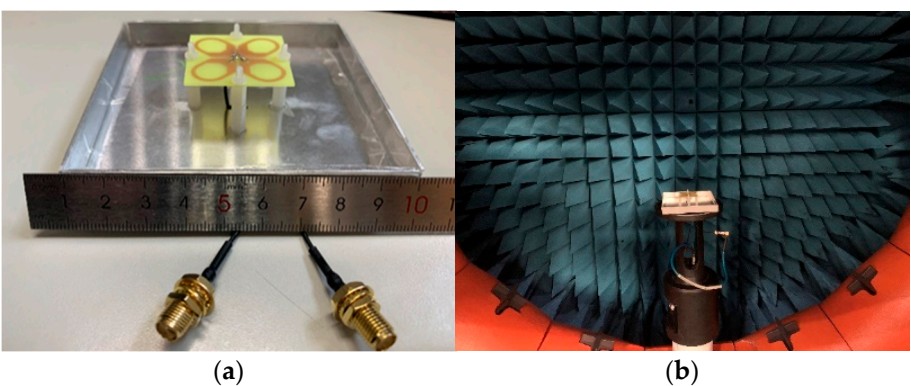

**Figure 9.** The prototype and the measured environment of the proposed antenna. (**a**) The prototype, (**b**) the measured environment of the proposed antenna.

The simulated and measured radiation patterns in E-plane and H-plane at 3.2 GHz, 4.2 GHz, and 5.2 GHz are displayed in Figure 11. The minimum simulated XPD is greater than 27 dB at 0° over the band, which is greater than 20 dB at ±60°. Due to the XPD below 4.2 GHz being greater than 40 dB, the cross-polarization radiation pattern cannot be seen in Figure 11a,b at the E-plane. Then, the front-to-back ratio (FBR) is greater than 21 dB. Additionally, the half power beam width (HPBW)

is 61° ± 3° at the E-plane and 68° ± 3° at the H-plane over the band, respectively, which shows the measured results are in good agreement with the simulated ones.

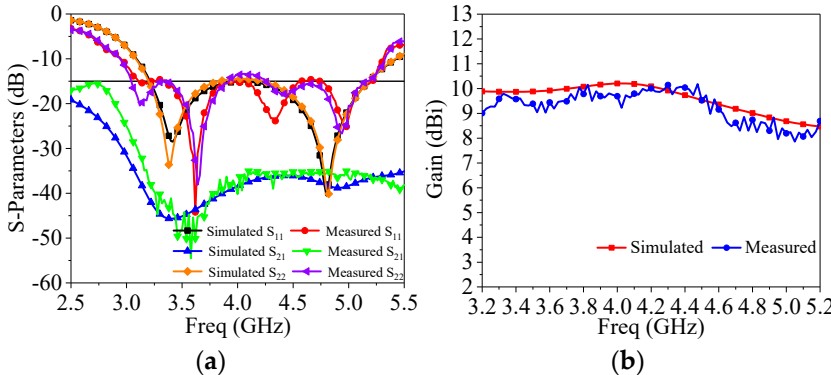

**Figure 10.** The simulated, measured S-parameters and gain. (**a**) S-parameters, (**b**) gain.

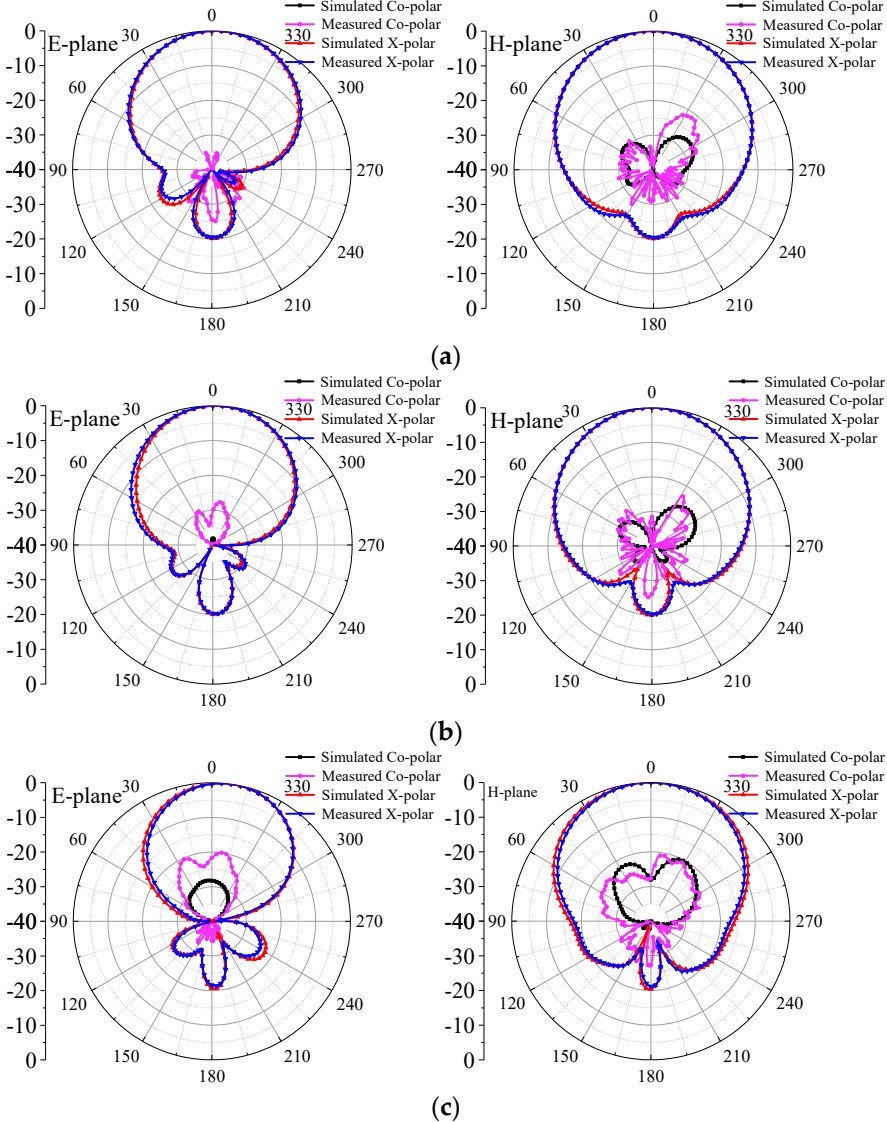

**Figure 11.** The Simulated and measured radiation patterns at different frequencies. (**a**) 3.2 GHz, (**b**) 4.2 GHz, (**c**) 5.2 GHz.

In order to demonstrate the performance of the proposed antenna, the comparison of the antenna with other recently base station antennas are listed in Table 2. Compared with them, higher isolation is achieved due to the introduction of suitable feeding structures. In addition, good matching enables the antenna to radiate more electromagnetic energy under the same conditions, so higher gain is achieved.

**Table 2.** Comparison of recent broadband antennas.

| Ref | Bandwidth (GHz) | Isolation (dB) | Gain (dBi) | XPD (0°, dB) | XPD (±60°, dB) | HPBW (degree) | Antenna Size (λ) |
| --- | --- | --- | --- | --- | --- | --- | --- |
| [18] | 1.7–2.7, 45% | >25 | 8.2 ± 0.6 | NG | NG | 68 ± 2 | 0.32 × 0.32 |
| [20] | 1.69–3.6, 72.2% | >25 | 8 ± 0.9 | NG | NG | 68 | 0.57 × 0.57 |
| [21] | 3.3–5.0, 41% | >32 | 8 | NG | NG | 70 ± 5 | 0.34 × 0.34 |
| [22] | 3.18–5.19, 48% | >30 | 8 | NG | NG | 75 | 0.49 × 0.49 |
| [23] | 2.21–3.63, 48.6% | ≥30 | 8.3 | ≥30 | NG | 65.5 ± 5.5 | 0.39 × 0.39 |
| Proposed | 3.2–5.22, 48% | >35 | 9.0 ± 1 | >27 | >20 | 68 ± 3 | 0.42 × 0.42 |

## 6. Conclusions

A broadband dual-polarized base station antenna covering 5G bands (sub-6 GHz band) with good performance is designed, manufactured, and measured. A wide band and good impedance matching are obtained by loading the Y-shaped feeding lines. In addition, the equivalent circuit model is introduced to verify the mechanism of the antenna, and the feasibility of the antenna structure is verified by comparing the simulation of the antenna structure and the antenna equivalent circuit. The simulated and measured results indicate that the high gain (average 9 dBi) and the stable radiation patterns (the HPBW is 61° ± 3° at E-plane and 68° ± 3° at H-plane) are achieved, respectively. Compared with other base station antennas, the proposed antenna obtains a wide band (48%, 3.2–5.22 GHz) and a higher isolation (35 dB). Additionally, the proposed antenna with simple structure and is convenient to be manufactured. Therefore, it can provide partial reference for the design of 5G base station antennas.

**Author Contributions:** Conceptualization: B.W., C.L.; methodology: C.L., C.Z.; validation: H.H., B.Y.; investigation: C.L., C.Z.; data curation: C.L.; writing—original draft preparation: C.L.; writing—review and editing: C.L. All authors have read and agreed to the published version of the manuscript.

**Funding:** This research was funded in part by the National Natural Science Foundation of China under grant no. 61601074 and in part by the Chongqing Technology Innovation and Application Development Project under grant no. cstc2019jscx-msxmX0049.

**Acknowledgments:** This work was supported in part by the National Natural Science Foundation of China under grant no. 61601074 and in part by the Chongqing Technology Innovation and Application Development Project under grant no. cstc2019jscx-msxmX0049.

**Conflicts of Interest:** The authors declare that there is no conflict of interests regarding the publication of this article.

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
