# Peer review of "Broadband Dual-Polarized Loop Cross-Dipole Antenna for 5G Base Station Applications"

_electronics, doi:10.3390/electronics9101574_

Round 1

Reviewer 1 Report

This manuscript presents a broadband dual-polarized antenna. The antenna is a loop cross-dipole. This reviewer does not find anything new in this manuscript. Even if there is some novelty in the design, it is not highlighted. The parametric study also does not show any significant change by changing different parameters. Therefore, this reviewer thinks that the manuscript is not in an acceptable form for publication. 

Reviewer 2 Report

Dear Authors,

Here are my comments:

*) Together with ref [4], the following comments is also suggested. Nested patches one top of the other are also used to have also multiband behaviour (ref)

ref1) C. Puente, J. Anguera, C. Borja, "Dual-band dual-polarized antenna array", US Pat. 6,937,206

ref2) J. Anguera, "Fractal and BroadBand Techniques for Miniature, Multifrequency and High-Directivity Microstrip Patch Antennas", Ph.D Dissertation at Universitat Politècnica de Catalunya, 2003

*) Discuss at the introduction, a littel bit about antenna array for 5G. For example, what gains do they have, beam steering, etc.

*) Figure 2, what happens if you dont have the hole on the dipole antenna?

*) Figure 3: please make xlabel and ylabel larger. Review also other figure labels

*) Figure 5, enlarge please, it is very small

*) Table 2, when compare, please do also incude the antenna area and heigh in terms of wavelength at the lowest frequency of operation

I hope these comments help to improve the paper.

Many thanks

Reviewer 3 Report

This paper presents a broadband Dual-Polarized Loop Cross-Dipole
3 Antenna for 5G Base Station Applications. The paper is well-written and technical sound, however, it still need improvement based on following comments:

1 - The abstract need to be rewritten, as it does not give adequate indicate with regards to the paper to readers. 

2 - The authors need to introduce a separate section section highlighting comparison of state of the art work.

 3- The conclusion part is also too short, please rewrite it properly.

4  - Mathematical modelling related to proposed work will help the readers.

5 - Contributions from each authors should be clearly indicated. Which part of the work was performed by each author.

Round 2

Reviewer 1 Report

This reviewer thanks the authors for their effort to enhance the quality of the manuscript. However, the manuscript needs more improvement. 

  1. How do the authors justify having two close resonances in the S-parameters? They showed a parametric study that may adjust the second resonance frequency by changing a parameter. How do they change the first resonant frequency? Which parameter can do it? They need to show this a another parametric study that the first resonance also is tunable.
  2. In Fig. 2, the authors have demonstrated that by removing an internal part of the radiator, the matching can be tuned. This however, needs to be clarified in two aspects. First, removing those part is mainly affect the value of matching. While it can be matched with the first design, by removing those parts, just a better matching is provided. The bandwidth does not change significantly. So, this cannot be considered a significant improvement, unless the authors mention a specific application where it needs a perfect matching. Second, did the authors study the effect of this on the antenna gain? They need to study this.
  3. How does this paper contribute to the 5G application? If this antenna could be incorporated with a larger structure for larger gain such as the ones presented in "Resonant Cavity Antennas for 5G Communication Systems: A Review" we could justify its application for 5G. Please read the above paper and justify how this antenna can be incorporated with FSS structure for 5G applications.
  4. It is again not clear how this antenna can be use in base stations. If the antenna could be used in the feed structure of resonant cavity structures, it would be a great candidate for base stations. However, this antenna alone is not of a great interest for base stations. To this reviewers, the authors should talk about the potential incorporation of their proposed structure with PRS structures in resonant cavity antennas that makes it appealing for base stations. I refer you to "Self Generation of Circular Polarization Using Compact Fabry–Perot Cavity Antennas", "Wideband High-Gain Circularly-Polarized Resonant Cavity Antenna with a Thin Complementary Partially Reflective Surface", and "Design of a Broadband Polarization-Reconfigurable Fabry–Perot Resonator Antenna". Please look at these papers to see how you can incorporate the structure with PRSs for the purpose of 5G base stations. The authors need to discuss this issue in the manuscript.

Reviewer 3 Report

The authors have addressed all my comments. I would recommend it for publication.

Author Response

We thank all the reviewers for thorough review, encouragement, constructive comments and corrections suggested.

Round 3

Reviewer 1 Report

The authors response to the reviewers' comments are appropriate. However, the authors need to apply them to the manuscript. This reviewer does not agree with the authors that the Base station reflection coefficient must be below -15 dB. In this regards, I think the novelty of this work is marginal. The title of the paper is about the base station that needs a high antenna gain. So this reviewer expect that at least address the comments raised in the previous submission. This reviewer even gave the authors a clue for 5G base stations by using resonant cavity antenna structure, and the authors did not address the comment in the manuscript. While they should talk about how their proposed antenna can be used in the structure of resonant cavity antenna for 5G BASE STATIONS (even some references were suggested), the authors tried to avoid talking about it in detail in the manuscript. The authors also added two irrelevant references to the manuscript.
